# Can Active Sampling Reduce Causal Confusion in Offline Reinforcement Learning?

**Gunshi Gupta***
University of Oxford

**Tim G. J. Rudner**
University of Oxford

**Rowan McAllister**
Toyota Research Institute, USA

**Adrien Gaidon**
Toyota Research Institute, USA

**Yarin Gal**
University of Oxford

## Abstract

Causal confusion is a phenomenon where an agent learns a policy that reflects imperfect spurious correlations in the data. Such a policy may falsely appear to be optimal during training if most of the training data contains such spurious correlations. This phenomenon is particularly pronounced in domains such as robotics, with potentially large gaps between open- and closed-loop performance of an agent. In such settings, causally confused models may appear to perform well according to open-loop metrics during training but fail catastrophically when deployed in the real world. In this paper, we investigate whether selectively sampling appropriate points from the dataset may enable offline RL agents to disambiguate the underlying causal mechanisms of the environment, alleviate causal confusion in offline reinforcement learning, and produce a safer model for deployment. To answer this question, we consider a set of tailored offline reinforcement learning datasets that exhibit causal ambiguity and assess the ability of active sampling techniques to reduce causal confusion at evaluation. We provide empirical evidence that uniform and active sampling techniques are able to consistently reduce causal confusion as training progresses and that active sampling is able to do so significantly more efficiently than uniform sampling.

## 1 Introduction

Offline learning offers opportunities to scale reinforcement learning to domains where offline data is plentiful but online interaction with the environment is costly. The fundamental challenge of offline reinforcement learning is to identify the cause and effect of actions from a fixed dataset, which is often intractable. In the absence of online interactions, our hope is that the dataset covers a uniform distribution of an exhaustive set of plausible scenarios. This is often not the case in datasets for robotic control, which are heavy-tailed and often contain only a handful of samples for rare (and informative) events. Causal confusion occurs when agents misinterpret the underlying causal mechanisms of the environment and erroneously associate certain actions or states with a given reward. For example, if an agent happens to simultaneously observe independent events $X$ and $Y$ in its environment whenever it receives a reward $R$, and $R$ causally depends on $Y$ but not on $X$, the agent may attribute the reward $R$ to $X$ and $Y$ occurring jointly even though $R$ may be independent of $Y$. Problematically, if the spurious correlation between $Y$ and $R$ observed at training time ceases to hold at deployment time, a causally-confused model may show a significant deterioration in performance.

Although spurious features might not *perfectly* explain all the input-label pairs in offline data, optimisation methods such as stochastic gradient descent benefit from correlations in the data when

---

*Correspondence to: `gunshi.gupta@lmh.ox.ac.uk`.

3rd Offline RL Workshop at the 36th Conference on Neural Information Processing Systems (NeurIPS 2022).

seeking to reduce the training loss. Therefore spurious correlations are unwittingly transferred from the data to the learned models. This phenomenon is especially pronounced in models trained on high-dimensional visual inputs, since extracting the true causal factors of an environment (and their interplay) from an image is a particularly difficult problem. On the other hand, it is noticeably easy for neural networks to find shortcuts for prediction, for instance, using co-occurrence patterns between objects and backgrounds to successfully perform object detection, as has been extensively documented in the deep computer vision literature [Beery et al., 2018, Geirhos et al., 2020]. Several works have also reported causal confusion in control policy learning and used heuristic loss re-weighting or dataset-balancing schemes based on domain knowledge of the prediction task to reduce the amount of causal confusion in the learned models [Wijmans et al., 2019, Kumar et al., 2022]. However, these heuristics require a practitioner to know the source of causal confusion a priori, since they require explicitly increasing the loss for specific spurious-correlation-breaking samples in the dataset [Wen et al., 2021, 2020]. While this may be limiting in practice, the success of these heuristics demonstrates that it is possible to recover less causally confused models from a fixed dataset, as long as we have access to an *oracle* sampling scheme for the training data.

Building on recent work that highlighted active sampling as a tool for sample-efficient causal estimation, we study uncertainty-based and loss-based active sampling techniques and find that active sampling alleviates causal confusion in offline training and yields higher rewards at evaluation. We further show that active sampling is able to alleviate causal confusion at a significantly higher sample efficiency than naive approaches, such as uniform sampling, and that the usefulness of active sampling in alleviating causal confusion is highly related to the quality of predictive uncertainty estimates used in the best-performing, uncertainty-based acquisition function. Our results demonstrate that while dominant spurious correlations in large offline datasets may lead to poor performance, and hence to harmful decision-making at deployment, active sampling techniques can significantly reduce the negative effects of spurious correlations on offline RL agents and that they are able to do so with high sample efficiency. The study conducted in this paper is located at the intersection between the fields of reinforcement learning, Bayesian active learning, and causal inference. In Sections 2 and 3, we provide relevant background on and connections between these three areas.

## 2   Preliminaries

We consider an environment formulated as a Markov Decision Process (MDP) $\mathcal{M}$ defined by a tuple $(\mathcal{S}, \mathcal{A}, \mathcal{P}, r, d_0, \gamma)$, where $\mathcal{S}$ is the state space, $\mathcal{A}$ is the action space, $\mathcal{P}(\mathbf{s}' \mid \mathbf{s}, \mathbf{a})$ is the transition probability distribution, $r : \mathcal{S} \times \mathcal{A} \to \mathbb{R}$ is the reward function, $d_0$ is the initial state distribution, and $\gamma \in (0, 1]$ is the discount factor. The goal of reinforcement learning (RL) is to find an optimal policy $\pi(\mathbf{a} \mid \mathbf{s})$ that maximizes the cumulative discounted reward $\mathbb{E}_{\mathbf{s}_t, \mathbf{a}_t} \left[ \sum_{t=0}^{\infty} \gamma^t r(\mathbf{s}_t, \mathbf{a}_t) \right]$, where $\mathbf{s}_0 \sim d_0(\cdot), \mathbf{a}_t \sim \pi(\cdot \mid \mathbf{s}_t)$, and $\mathbf{s}_{t+1} \sim \mathcal{P}(\cdot \mid \mathbf{s}_t, \mathbf{a}_t)$.

$Q$-learning-based RL algorithms learn an optimal state–action value function $Q^*(s, a)$, representing the expected cumulative discounted reward starting from $s$ with action $a$ and then acting optimally according to policy $\pi^*$ thereafter, ie $Q^*(s, a) = \mathbb{E}_{\pi^*} \left[ \sum_{t=0}^{\infty} \gamma^t r(\mathbf{s}_t, \mathbf{a}_t) \mid s_0 = s, a_0 = a \right]$. Analogously, the value function $V(s) = \mathbb{E}_{\pi} \left[ \sum_{t=0}^{\infty} \gamma^t r(\mathbf{s}_t, \mathbf{a}_t) \mid s_0 = s \right]$ represents the expected cumulative discounted reward achievable from state $s$ when following policy $\pi$. $Q$-learning is trained on the Bellman equation defined as follows with the Bellman optimal operator $\mathcal{B}$ defined by:

$$\mathcal{B}Q(s, a) := R(s, a) + \gamma \mathbb{E}_{P(s' \mid s, a)} \left[ \max_{a'} Q(s', a') \right]. \tag{1}$$

The $Q$-function is updated by minimizing the Bellman Squared Error $\mathbb{E}\left[ (Q - \mathcal{B}Q)^2 \right]$ where a frozen, periodically updated copy of the $Q$-network weights are used to compute the target $\mathcal{B}Q$. Offline reinforcement learning algorithms aim to learn an optimal policy by learning estimates of the value (or $Q$-value) function from a static dataset of transitions $\mathcal{D} = \{(\mathbf{s}, \mathbf{a}, r, \mathbf{s}')\}$ collected by a behaviour policy $\pi_\beta$. Since the agent does not interact with the environment, naively using online RL techniques in the offline case leads to value function overestimation on unseen states and actions [Hasselt, 2010]. Thus, algorithms based on the *pessimism principle* of underestimating the $Q$-values to optimise the worst-case regret bound have been successful at learning policies from datasets containing either good coverage of the state–action space or high return trajectories [Kumar et al., 2020, Kumar et al., 2019, Buckman et al., 2021, Bai et al., 2022]. Prior work categorised pessimistic offline RL algorithms into (1) *proximal* and (2) *uncertainty-aware* algorithms, where the former penalise actions not seen in the data and the latter construct value function updates taking the uncertainty of the targets $\mathcal{B}Q$ into account [Buckman et al., 2021]. Proximal pessimistic algorithms like CQL, BEAR [Kumar et al.,

2019, Kumar et al., 2020] are known to work well with exactly the kind of narrow and biased data distributions that are most prone to causal confusion [Fu et al., 2020]. We, therefore, analyse the proximal class of algorithms in this work and leave the study of causal confusion in uncertainty-aware pessimistic offline RL algorithms to future work.

**Conservative $Q$-Learning.** We choose CQL [Kumar et al., 2020] as an instantiation of a proximal pessimistic offline RL algorithm in our experiments, due to its simplicity and competitive performance on offline RL benchmarks. The CQL objective, which combines the standard TD-error of $Q$-learning with a penalty constraining deviations from the behaviour policy, is defined as:

$$\mathcal{L}_{\text{critic}}^{\text{CQL}}(\theta) = \frac{1}{2} \mathop{\mathbb{E}}_{(s,a,s')\sim\mathcal{D}} \left[ (Q_\theta(s,a) - \mathcal{B}Q_{\bar{\theta}}(s,a))^2 \right] + \alpha_0 \mathop{\mathbb{E}}_{s\sim\mathcal{D}} \left[ \log \sum_a \exp Q_\theta(s,a) - \mathop{\mathbb{E}}_{a\sim\pi_\beta}[Q_\theta(s,a)] \right],$$
(2)

where $\alpha_0$ controls the degree of conservatism and $\theta$ is the parameterisation of the learned $Q$-function.

## 3 Alleviating Causal Confusion in Offline RL via Active Sampling

In this section, we briefly describe the active sampling schemes we use in this work in conjunction with offline RL. In Appendix C, we define our assumptions on long-tailed datasets that may lead to causally confused models, and in Appendix D, we discuss how active sampling can be seen as facilitating causal discovery in offline (observational) data.

### 3.1 Active Sampling

Active sampling techniques are used to selectively sample from a given dataset during training to enable sample-efficient learning or to improve learning from noisy data [Loshchilov and Hutter, 2015, Mirzasoleiman et al., 2019]. To perform active sampling, data points are scored based on a given information-theoretic acquisition function, and a small number of points are sampled based on these scores using a pre-specified weighting scheme. In this work, we build on Jesson et al. [2021] and study whether active sampling can enable sample-efficient resolution of causal confusion in models trained on long-tailed datasets . In particular, the focus of this work is to investigate whether active sampling techniques can be used to address the challenges of causal ambiguity in state–action trajectories used in offline RL —without necessitating any modifications to the learning objective. Algorithm 1 in the Appendix describes the CQL algorithm with active sampling of transitions, where the modifications from uniform sampling are highlighted in blue. We study the following uncertainty-based and loss-based data acquisition functions:

**Uncertainty-based Sampling.** We are interested in sampling state transitions for which the learned $Q$-network is uncertain about its predictions over the action seen in the data ($a_\beta = \pi_\beta(s)$) or its own greedy action ($a^* = \arg\max Q(s)$). We model the epistemic uncertainty in the learned $Q$-function, by creating an ensemble model of $Q$-functions $\{Q_{\theta^i}\}_i$ and training on identical transitions across the ensemble members, with their own corresponding targets, $\{\bar{\theta}^i\}_i$, as proposed in Ghasemipour et al. [2022]. We can then use the variance of an estimate across the ensemble as the measure of the epistemic uncertainty about the estimate. However, the $Q$-values of different ensemble members could have arbitrary numerical offsets but still be equivalent, since they're trained by bootstrapping value estimates. Instead, we estimate the uncertainty about an action by computing the variance of its advantage estimates over the ensemble, where the advantage of action $a^i$ for a $Q$-learner is given by

$$A^\pi(s, a^i) = Q^\pi(s, a^i) - V^\pi(s) \approx Q^\pi(s, a^i) - \sum_a \left[ Q(s,a) \cdot \frac{e^{Q(s,a)}}{\sum_{a'} e^{Q(s,a')}} \right]$$
(3)

The advantage function represents a causal quantity assessing the relative *effect* of action $a^i$ on the outcome $Q$ for a given state $s$. The acquisition scores in this case can then be computed by the expression: $\text{Var}(A^\pi(s, a^i))$. We consider two variants, *Variance-greedy* and *Variance-data*, referring to whether the advantages for the greedy ($a^i = a^*$) or the dataset actions ($a^i = a^\beta$) are computed.

**Loss-based Sampling.** We sample transitions based on their Temporal Difference error similar to Prioritised Experience Replay (PER; Schaul et al. [2015]) and refer to this variant as *TD-Error*.

We provide further implementation-related details about active sampling in Appendix E.1.

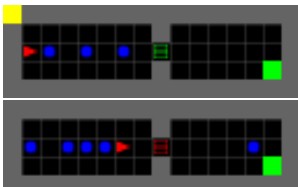

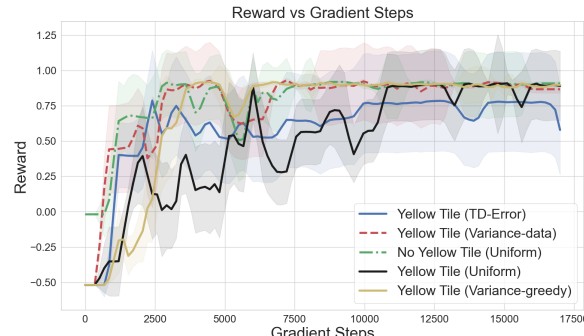

(a) The agent, other vehicles and the goal are depicted by the red triangle, blue circles and green square respectively. The traffic light is the striped square in the middle of the grid. **Top:** The top-left tile flashes yellow since the leading vehicle blocking the agent is static. **Bottom:** The agent is in front of a red light, and the top-left tile isn't yellow since the leading vehicle isn't static or blocked.

(b) **Evaluation Reward:** Uniform sampling takes $4\times$ gradient steps to recover the correct solution compared to active sampling (*TD-Error*, *Variance-greedy* and *Variance-data*) when both are trained on data with the spurious yellow tile.

Figure 1: Traffic-world environment.

# 4 Experiments

In the following section and in Appendix G, we describe our experiments on three benchmark environments investigating the following questions: **(1)** Can causal confusion be consistently observed in offline RL agents when sampling transitions uniformly from a long-tailed demonstration dataset? **(2)** Does active sampling based on a model's predictive uncertainty or its loss help in alleviating causal confusion? **(3)** To what extent does the quality of predictive uncertainty estimates affect predictive performance and sample efficiency gains under uncertainty-based active sampling? (Appendix G.3)

## 4.1 Illustrative Example: Traffic-World

We build on the environment proposed in [Anonymous, 2021] to construct a gridworld (shown in Figure 1a), where an agent starting at the leftmost point behind leading vehicles, needs to cross a traffic light to reach the goal on the bottom right. We collect data such that the probability of the traffic light turning red becomes lower as the agent approaches it, and so the data distribution contains: **(1)** mostly episodes where the light is green throughout the episode, **(2)** some episodes where the traffic light is red and the agent has to wait behind the vehicle in front (referenced here onward as the leading vehicle) before the light turns green again, and **(3)** only a couple of episodes where the light turns red with the agent at the front of the traffic queue. To test causal confusion explicitly here, we introduce a related spurious correlate: a flashing yellow tile at the top left of the grid (emulating the brake lights on a leading vehicle), that is yellow whenever the leading vehicle is stopped or blocked, and grey otherwise. The agent could simply follow this as an indicator of whether to stop or go ahead, and this policy would be correct for 98% of the data points. We report the average reward over three instantiations of this environment testing different types of causal confusion. Figure 1b shows the evaluation curves of CQL agents trained with uniformly-sampled data, with and without the yellow tile present in images in the dataset. We see that the performance of the former agent degrades significantly compared to the latter. Also shown are the active sampling variants trained with the spurious yellow tile, which perform very similarly to uniform sampling without the spurious correlate present. Appendix F provides further details and analysis related to this experiment.

# 5 Conclusion

In this paper, we studied how to alleviate causal confusion in offline RL. We designed uncertainty- and loss-based data sampling baselines to selectively sample transitions for training, and found evidence that active sampling can recover a less causally-confused model in significantly fewer training steps as compared to uniform sampling. In future work, we hope to scale the analysis performed in this paper to larger benchmark environments, with sources of noise in the reward and transitions, to further corroborate our findings. Such an analysis would help further distinguish the quality of solutions found through loss-based and uncertainty-based active sampling since noisy transitions can have *irreducible* loss. Additional promising avenues for future work include extending the current analysis to settings with continuous action spaces.

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

# A   Appendix

# B   Related Work

**Causal Confusion in Supervised Policy Learning.**   Several works in offline imitation learning have proposed solutions to mitigate causal confusion, which was first defined by de Haan et al. [2019]. The general practice across these works is to select sequential-decision-making benchmarks like simulated robotic control, augment their inputs with nuisance variables to induce spurious correlations, and show that policies trained on this data are inferior to those trained without the nuisance variables in the data. de Haan et al. [2019] demonstrated causal mis-identification in models trained in this manner on expert trajectories collected in the Mountain-Car and the CARLA simulators [Sutton and Barto, 2018, Dosovitskiy et al., 2017] where inputs were augmented with the previous control command taken by the acting agent. They proposed to resolve the confusion through a scheme to query an expert or collect additional rollouts in the environment to refine a learned causal graph that conditions the learned policy. Wen et al. [2020] proposes adversarial training to prune out any *known* sources of spurious correlations from the policy's representation, for instance, the previous control commands given to a robot; Wen et al. [2021] propose re-weighting the losses of data points based on the loss of a model trained with just the spurious correlates as the input; OREO [Park et al., 2021] regularises the model's representation to be invariant to any individual object being dropped out in a scene. Lee et al. [2022] propose training a diversified policy ensemble for imitation learning in the case when *perfect spurious correlations* exist in the data and later select from these hypotheses based on validation data Causal Confusion has also recently been studied in reward-learning from preferences [Tien et al., 2022], where spurious correlations can be drawn between a human evaluator's preferences and certain actions or parts of the state space, for tasks in the Assistive gym [Erickson et al., 2019]. For instance, a reward model trained to classify or rank trajectories for a feeding task can easily learn that higher force correlates to higher reward since a certain level of force needs to be inserted in the direction of the mouth in all preferred trajectories.

**Active Sampling.**   Our work is inspired by the work of Jesson et al. [2021], which looks at the problem of treatment effect estimation in settings where we want to be sample-efficient in terms of querying for outcomes of costly experiments. They propose several causality-inspired acquisition functions that prefer data-points which have both high variance in their estimated outcomes and correspond to covariates with considerable overlap in the dataset. In $Q$-learning-based RL algorithms, Prioritised replay [Schaul et al., 2015] is a non-Bayesian loss-based sampling scheme proposed for off-policy learning. It computes acquisition scores based on the TD-error of transitions and has not been studied in offline RL.

**Ensemble Models in RL.**   Ensembles have been studied extensively to guide exploration in online RL [Osband et al., 2016, Lee et al., 2021], and recently to construct adaptive pessimism constraints in offline RL, to disincentives uncertain actions from having high estimated returns. It was recently also shown that significantly increasing the size and diversity of the ensembled critic in Soft-Actor-Critic [Haarnoja et al., 2018] performs competitively with state-of-the-art offline RL algorithms [An et al., 2021]. We are not aware of any prior work that has explored how the uncertainty of the value function could be used to sample transitions in RL.

**AI Alignment.**   AI alignment seeks to align the behavior of agents with the intentions of their creators by investigating the incentives behind demonstrated tasks. Recent work on *Goal Mis-generalisation* [Langosco et al., 2022] explores how online RL agents in Procgen [Cobbe et al., 2019] can get confused about the goal they are pursuing since those goals co-occur with irrelevant artifacts in the environment most of the time. In this case, the specification is correct, but the agent still pursues an unintended objective (as opposed to poor reward definitions that predictably lead to reward hacking). We build upon an environment introduced in this work to collect datasets to reproduce the phenomenon of causal confusion in offline RL.

# C   Setup: Causally Ambiguous State–Action Trajectories

We adopt the terminology of de Haan et al. [2019] and partition the fully observable state space $\mathcal{S}$ into $n$ factors, composed of $m$ causal factors which are causal parents of the optimal state–action value

function $Q^*(S, \cdot)$, and $n - m$ nuisance variables which do not influence $Q^*(S, \cdot)$: $S = [S_1, S_2, ..., S_m, S_{m+1},...,S_n]$. Note that in this section, the superscript denotes the index of a point in the dataset ($t$) whereas the subscript denotes the index of the latent causal factor. In this case, the following conditional independence relation indicating a lack of causal influence of $S_{m+1:n}$ on $Q(S, \cdot)$ should hold for every state $s$ and action $a$:

$$p(Q^*(s, a) = q \mid s_{1:m}, s_{m+1:n}) = p(Q^*(s, a) = q \mid s_{1:m}). \tag{4}$$

We assume that the offline data $\mathcal{D}$ is distributed such that $S_{m+1:n}$ is highly correlated with $Q(S, \cdot)$, and so the independence relation does not hold when the function $Q$ is statistically estimated from transitions in $\mathcal{D}$. Regardless of the full observability of all the true causal factors in the input $S^t$, several works including that of Shah et al. [2020] have shown that a *simplicity bias* in stochastic gradient descent makes it likely for the learned $Q$-function to use $S_{m+1:n}$ as a predictive feature. This becomes more likely as the correlation between $S_{m+1:n}$ and $Q(S, \cdot)$ becomes stronger in the dataset.

In our experiments, we simulate the presence of the non-causal factors $S^t_{m+1:n}$ by augmenting the dataset with minimal nuisance correlations and verify that agents trained with these additions exhibit a significant decrease in performance at evaluation. We ascribe this deterioration in performance to the model choosing to attend to the spurious correlate and mapping two meaningfully distinct states $s^1$ and $s^2$ (where $\arg\max(Q^*(s^1, \cdot)) \neq \arg\max(Q^*(s^2, \cdot))$) to a similar distribution of $Q$-values. We see this during qualitative evaluation in our benchmark environments, which we design to test the influence of the spurious correlate on the learned policy's performance. Notably, since $\mathcal{D}$ does contain (a potentially small number of) datapoints that break the dominant spurious correlations, the challenge in this setting is subtly different from the broader problem of achieving good generalisation, where the goal is to learn state abstractions that lead to accurate $Q$-values on new states with small unseen variations around concepts represented in the training data. In our setting where the long-tailed $\mathcal{D}$ is composed of a large number of causally-ambiguous trajectories, we want a model to learn state abstractions that at the very least explain all of the training data and not only the modes of the distribution of scenarios represented in $\mathcal{D}$. In Appendix D.1, we describe the connection between $Q$-values (and derived quantities like the advantage function) in RL and quantities of interest in the field of causal inference. This will elucidate why we chose active sampling as a tool for causal discovery in the following section.

# D   Causal Inference

## D.1   Estimating the Conditional Average Treatment Effect from Causally Ambiguous Data

Treatment-effect estimation, where the goal is to estimate the effect of a treatment $T \in \mathcal{T}$ on the outcome $Y \in \mathcal{Y}$ for individuals described by covariates $X \in \mathcal{X}$, is a central problem in causal inference. In particular, we may wish to estimate the expected difference in potential outcomes for individuals when subjected ($t = 1$) or not subjected ($t = 0$) to a treatment $t$, measured by the Conditional Average Treatment Effect (CATE, Abrevaya et al. [2015]), defined as

$$\tau(X) \equiv \mathbb{E}[Y \mid x, t = 1] - \mathbb{E}[Y \mid x, t = 0]. \tag{5}$$

Here the treatment $t$ is often considered to be a binary variable, but the definition can be extended to the multivariate case with continuous or discrete values. Realizations of the random variables $X, T, Y$ are denoted by $x, t, y$, respectively. We list the full set of assumptions needed to ensure identifiability of the CATE estimator in Appendix D.2.

To frame the problem of disambiguating the effect of different actions (i.e., treatments $t \in T$) in a given state (i.e., a set of covariates $x \in \mathcal{X}$), we frame estimation of the advantage functions $A(s, a) = Q(s, a) - V(s)$ in reinforcement learning as CATE estimation [Pan et al., 2021]. In particular, the outcome $Y$ corresponds to the $Q$-function (i.e., the expected return) for a given state–action pair, and we can express the corresponding CATE estimator as

$$\mathbb{E}\left[\sum_{t=0}^{\infty} \gamma^t r\left(\mathbf{s}_t, \mathbf{a}_t\right) \mid s_t = s, a_t = a\right] - \mathbb{E}\left[\sum_{t=0}^{\infty} \gamma^t r\left(\mathbf{s}_t, \mathbf{a}_t\right) \mid s_t = s\right] = Q(s, a) - V(s) = A(s, a), \tag{6}$$

which indicates the advantage of executing an action $a$ at state $s$ as opposed to any other alternative action. With this connection in established, we can now cast active sampling of data points used to estimate a value function as a sequential process of accurately estimating treatment effects.

## D.2 Assumptions in Causal Inference

In this section we list out the assumptions adopted within the potential outcomes framework defined in the Rubin-Neyman causal model [Rubin, 1974, Splawa-Neyman et al., 1990], to ensure *identifiability* of the CATE estimator.

**Assumption 2.1. (Consistency)** $y = ty^t + (1-t)y^{1-t}$, i.e. an individual's observed outcome $y$ given assigned treatment $t$ is identical to their potential outcome $y^t$.

**Assumption 2.2. (Unconfoundedness)** $(Y^0, Y^1) \perp T \mid \mathbf{X}$.

**Assumption 2.3. (Overlap)** $0 < \pi_t(\mathbf{x}) < 1 : \forall t \in \mathcal{T}$, where $\pi_t(\mathbf{x}) \equiv P(T = t \mid \mathbf{X} = \mathbf{x})$ is the propensity for treatment for individuals described by covariates $\mathbf{X} = \mathbf{x}$. When these assumptions are satisfied, $\widehat{\tau}(\mathbf{x}) \equiv \mathbb{E}[Y \mid T = 1, \mathbf{X} = \mathbf{x}] - \mathbb{E}[Y \mid T = 0, \mathbf{X} = \mathbf{x}]$ is an unbiased estimator of $\tau(\mathbf{x})$ and is identifiable from observational data.

In practice, one or more of these assumptions are relaxed because observational data that is collected for safety-critical tasks has very little scope for randomisation of a behaviour policy, indicating limited overlap.

---

**Algorithm 1** Conservative $Q$-Learning ( + active sampling)

---
1: Initialise ensemble $Q$-function $Q_\theta$, $n_{ep}$=epochs, $d_{sz}$=dataset size, $b_{sz}$=batch size, $T$=steps-per-epoch.
2: **for** epoch $e$ in $\{1, \ldots, n_{ep}\}$ **do**
3:     **for** step $t$ in $\{1, \ldots, T\}$ **do**
4:         compute scores $acq_i$ over $\mathcal{D}_{\text{train}} = [s_i, a_i]_{i=1}^{d_{sz}}$ according to the acquisition function
5:         $acq_i = \frac{acq_i}{\sum_{j=1}^{d_{sz}} acq_j}$ (normalise scores)
6:         sample batch $B = [s_i, a_i, s_i', r_i]_{i=1}^{b_{sz}}$ from $\mathcal{D}_{\text{train}} \sim multinomial(acq)$
7:         Train the $Q$-function on $D_{train}$ using objective from Equation (2)
8:     **end for**
9: **end for**

---

# E Implementation

All our environments use a discrete action space. Therefore we build our method on top of the double-DQN implementation similar to the original CQL paper. As stated in Section 3.1, we use ensembles of $Q$-networks where the ensemble members are trained with independent targets. At evaluation time, we average the $Q$-value outputs of the ensemble and select the action with the maximum $Q$-value. At other points during training where we need to do inference (for instance: to compute $Q$-values for computation of the conservative loss), we take the average across the ensemble as the $Q$-value outputs, similar to evaluation time.

A design choice we make is the network initialisation used when we start to do active sampling: the uncertainty encoded by a randomly initialised network at epoch 0 can be very inaccurate and biased towards some subset of the data. We instead train with uniform sampling for a single epoch and then start active sampling second epoch onwards. We also observe that bigger ensembles encode better measures of uncertainty than smaller ones when the network is trained for a lesser period than one epoch with uniform sampling at the start (Figure 6).

## E.1 Computing Acquisition Scores

Algorithm 1 (line 4) indicates that acquisition scores are computed for data points in $\mathcal{D}_{\text{train}}$ before every training step. However, in practice, computing the scores over all the transitions in the dataset can be both expensive and redundant since high-error or high-uncertainty points will likely stay high over a short window of subsequent gradient steps. Thus, we only recompute all the scores after every $n$ gradient steps and vary $n$ as a hyper-parameter in our experiments.

In Appendix G, we also explore a scheme where scores are recomputed only over-sampled batches, as in [Schaul et al., 2015]. This is an approximation where a priority queue is maintained with priorities

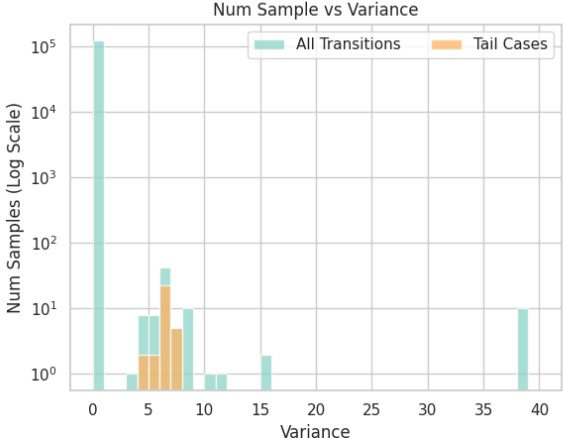

Figure 2: (**Traffic-World**) Histogram of the advantage function variance estimates for the data action plotted for all transitions, overlayed by the scores for one tail case's samples. The tail case we consider here comprises state-action pairs where the top-left tile flashes yellow but the data-collecting policy still moves one step ahead ignoring the tile, since there is space to move towards the goal without getting a negative penalty or triggering an episode termination. We can see that these samples fall in the 98th percentile of all points.

of every data point deriving from the acquisition score computed on it. In this case, the priorities are only updated for a small subset of points at every gradient step, since the acquisition scores are only recomputed on the data points in the sampled batch, potentially leading to many points in the replay buffer with *stale* scores. We adopt the same implementation as that of PER Schaul et al. [2015] with the scores for the priorities coming from the TD-error and Variance estimates. We include further details about this procedure in the appendix. We refer to the above two cases by appending *-dataset* and *-batch* to the names of the sampling schemes to indicate that scores are recomputed for the entire data versus just the sampled batch respectively.

## F    Traffic-World Experiment: Additional Details

In the traffic-world environment with the spurious correlate as the yellow tile, the following are two examples of tail-case scenarios in the corresponding collected dataset: (1) When the tile flashes yellow because the leading vehicle is static but the agent is not blocked because there is space to move ahead behind the leading vehicle, and (2) When the leading vehicle has crossed the traffic light which has just turned red in front of the agent, and the top-left tile is not flashing yellow because the leading vehicle is not static. However, if the agent takes the tile being *off* as a cue, it will decide to move forward and end up breaking the red light and fetching a negative reward. In Figure 2, we plot a histogram for the computed variance scores over the data–action's advantage estimates from the model (*Variance-data* scores) after one epoch of training with uniform sampling. We can verify that samples from the tail-case fall in the higher percentile of variance scores.

## G    Additional Results

### G.1    Assessing Generalization in Offline Reinforcement Learning: Procgen

The Maze environment in Procgen [Cobbe et al., 2019] defines a navigation task where the agent starts at the bottom left in the maze and receives a reward of +10 upon successfully reaching the goal which is sampled at any valid location in the maze. Langosco et al. [2022] recently showed that an agent trained on a series of environments with the goal always at the top-right will be causally confused about the source of the reward and will still navigate to the top-right even when the goal is sampled elsewhere. We generate a skewed *mixture* dataset containing mostly episodes where the goal is sampled at the top-right and a few episodes where the goal is sampled randomly. Further details about the setup are described in the Appendix. Figure 3b shows the evaluation performance of random and active sampling agents trained on the *mixture* dataset when goals are sampled randomly in the evaluation environment. We plot the computation time for the uniform and active sampling

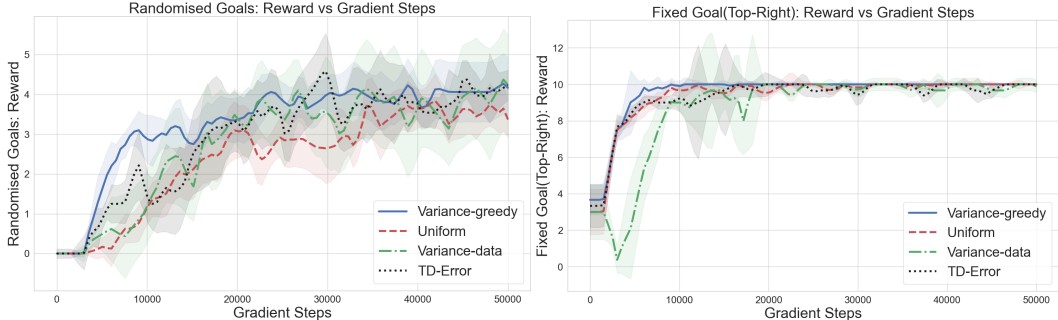

(a) Performance with randomly-sampled goals.  (b) Performance with goals at the top-right.

Figure 3: Agents trained on a dataset containing 6000 episodes with a fixed goal and 200 episodes with a randomly sampled goal in Maze. We see that uniform sampling and active sampling perform similarly in the fixed goal evaluation environment (right), but the active sampling variants achieve higher rewards in the environments with randomly sampled goals. This verifies that the model is not just performing well in one of the two kinds of environments, is not constrained by capacity, and the reason behind the lower performance of uniform sampling, in this case, is simply causal confusion.

variants in Figure 8 in the appendix and note that variance-based sampling reaches the highest score achieved by uniform sampling in lesser wall-clock time. Qualitative evaluations show that uniform sampling agents which achieve a lower reward still successfully navigate to the top-right corner of *hard* mazes, and are therefore only confused about the location of the goal.

It is important to note that if the evaluation environments considered here had followed a distribution of scenarios similar to the training data distribution, then we would have observed an evaluation curve similar to the weighted sum of the two curves in the left and right panels of Figure 5. The weight of the curve on the right would contribute significantly more since those scenarios (fixed top-right goals) would occur much more often in the evaluation environment. We would conclude that performance has saturated and the model has converged somewhere between 10,000–20,000 gradient steps since that is when the evaluation performance on the environment with fixed goals saturates. An active sampling model that was early-stopped at this point, would then fare much better on the tail cases (and do similarly otherwise) than the uniformly sampled model. This also motivates the design of our experiment setup where we define a set of deterministic evaluation environments for each scenario separately to enable model evaluation on an equally hard set of environments across seeds and methods and capture performance on both the head and tail of the distribution of scenarios.

### G.2 Causal Confusion in the ALE Benchmark

We are now interested in evaluating our active sampling baselines on a larger pre-existing dataset with more realistic noise and variations. Prior work in imitation learning [Park et al., 2021, de Haan et al., 2019] has attempted to simulate causal confusion in specific Atari game-play datasets by modifying images to display the previous action taken by the agent. This kind of causal confusion is inspired by robotics datasets that have trajectories with highly correlated (and predictable) sequences of actions because embodiment dictates that an agent's state does not change too drastically between subsequent timesteps.

However, when we trained uniform sampling CQL agents with and without the previous action on the images for 20 Atari games, only a select few games actually exhibited causal confusion of this kind. We pick Enduro since we can consistently observe both a convergence speed degradation and final reward degradation upon adding the previous action to the image. Enduro is a car racing game where an agent needs to overtake cars (receiving a positive reward for each car overtaken) and drive along a winding road, with a limited number of collisions allowed.

Figure 4 shows our proposed active sampling baselines compared to uniform sampling in the following two cases that we previously discussed in Appendix E.1: (1) *-dataset* case when we recompute the scores across the dataset every few gradient steps and (2) *-batch* case when the scores are only recomputed for the sampled batch and updated in a priority queue. Also shown is the uniform sampling baseline trained without the previous action displayed on the image (data without causal

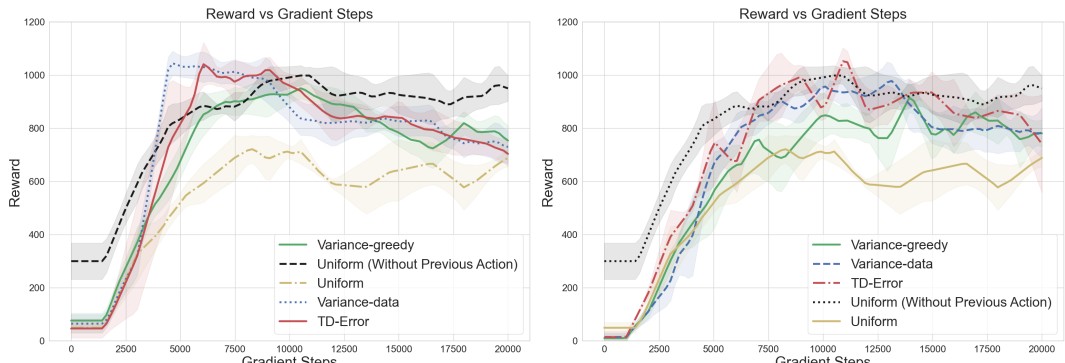

Figure 4: Comparison of active and uniform sampling agents on Enduro. On the left is the comparison in the *-dataset* case, and on the right is the case with *-batch* sampling where scores are incremented and updated based on the sampled batch. We see an increase in the number of gradient steps it takes *-batch* case to reach the highest possible reward.

ambiguity). We note that all of the active sampling variants perform significantly better than their uniform sampling counterpart not just in terms of convergence to their best performance but also in terms of the highest reward achieved.

### G.3  Effect of Predictive Uncertainty on Active Sampling and Causal Confusion

To better understand the improvement in convergence speed of active sampling vis-'a-vis uniform sampling, we train CQL with a smaller ($n = 3$) and a larger ($n = 10$) ensemble of $Q$-networks, all else equal, and sample identical transitions based on the predictive uncertainty of only either the smaller or the larger ensemble. This experiment setup allows us to isolate the effect of the predictive variance on active sampling and therefore on the observed reduction in causal confusion.

Figure 5 shows the training curves for both ensembles, and we find that when transitions are sampled according to the predictive uncertainty of the larger ensemble, even the smaller ensemble converges faster (albeit to a smaller value than that achieved by the larger ensemble). Similarly, when points are sampled according to the predictive uncertainty of the smaller ensemble, the larger ensemble converges to its highest reward more slowly.

This result implies that improved predictive uncertainty estimation, for which we use the size of the ensemble as a proxy, improves the ability of active sampling to identify samples that break spurious correlations in the data and reduce causal confusion. Notably, even when training CQL using the smaller ensemble, which we expect to produce worse mean $Q$-function estimates than the medium-size ensemble, using the larger ensemble's predictive uncertainty for active sampling results in a higher reward than for the medium-size ensemble, implying that the positive impact of improved predictive uncertainty estimates outweighs the adverse impact of the reduction in accuracy of the $Q$-function estimator. We provide a further discussion of these observations in Appendix I.

## H  Evaluation Metrics

We present the reward curves directly in most of our experimental reporting. The curves are computed by taking the inter-quartile-mean (IQM) across seeds as proposed by Agarwal et al. [2021]. We report the number of seeds used for all experiments in Appendix J.

Offline RL training performance is known to be non-monotonic, unlike supervised learning where the accuracy (or loss) increases (decreases) and then saturates. Often the reward curves start to decrease after a period when overfitting to the data-action values (through the conservatism penalty) starts to happen. Since we define deterministic benchmark environments testing all the tail scenarios in the case of Traffic-World and Maze experiments, we can consider any point on the curve where the model solves the highest number of environments (achieves the highest reward) as the point of convergence (as opposed to it being a noisy spike due to stochastic evaluation). This is similar to taking the max-reward checkpoint as done in [Agarwal et al., 2020]. However, to ensure that the solutions learned by any method are recoverable, we want to be able to get a good checkpoint from the model

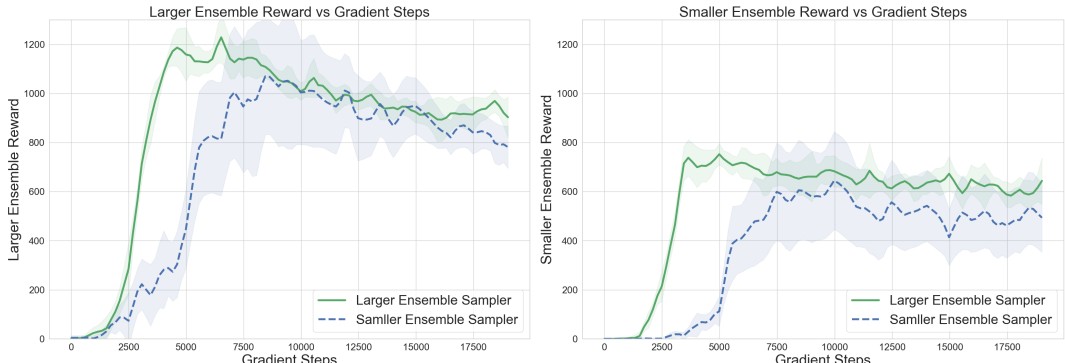

Figure 5: Reward curves of the larger (left) and smaller (right) ensemble training when sampling on the basis of the uncertainty of either one: of the speed of convergence depending on the quality of uncertainty estimation.

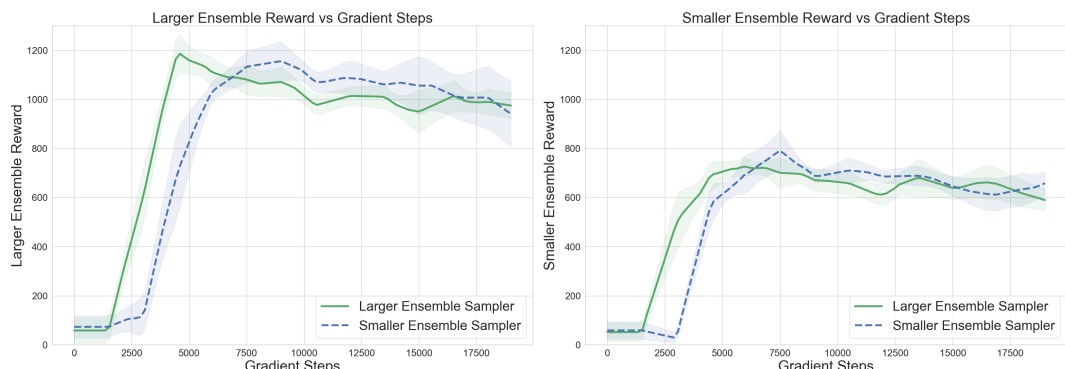

Figure 6: Reward curves of the larger (left) and smaller (right) ensemble training when sampling on the basis of the uncertainty of either one: of the speed of convergence depending on the quality of uncertainty estimation. Here, the difference is that both ensembles start active sampling from an initialisation that is trained for longer with uniform sampling. We see a less drastic gap in convergence between the two samplers, as compared to the main paper's Figure 5. This is expected since the uncertainties should get more informative (up to a point) as we train for longer with random sampling.

without needing to evaluate it too often. Thus we consider a method as having achieved higher reward than other methods at any point in time if it maintains this gap for at least two subsequent post-epoch evaluations. This is akin to taking a windowed-max over the reward curves.

## H.1 Code and Data

We will release our code, data and pretrained models once the work is uploaded online. The code repository will also contain code to reproduce all the figures in this work.

## H.2 Data Collection

1. Traffic-World: To collect data for Offline RL, we trained a PPO agent on a slightly modified version of the Traffic-world environment, with reward shaping on the environment [Anonymous, 2021], to incentivise the agent to reach the goal since this is a hard exploration environment (there is the potential to receive many negative rewards before receiving a positive reward, and without reward shaping the PPO agent just learns to toggle in-place till the episode ends to avoid negative penalties).

2. Maze: We use the Impala-based PPO agent trained in [Langosco et al., 2022] for 200M steps to collect the expert trajectories on 6000 episodes of episodes with randomised goals and 200 episodes of episodes with fixed goals.

# I  Further Analysis

In addition to the figure in the main paper, we plot in Figure 6 the training curves of two ensembles when they've been initialised with a longer period of uniform sampling at the start. We see that the gap between the informative-ness of samples fetched based on their uncertainties reduces (as compared to Figure 5). The larger ensemble's uncertainty still converges quicker, however, than when both ensembles are training using the smaller ensemble's uncertainty for data sampling.

One observation we made was related to active sampling results in the case of benchmarks with short episodes ($n_{steps} \sim 20 - 30$) and sparse rewards, in our case Traffic-World and Maze. $Q$-learning is trained through bootstrapping where we minimise the TD-error which involves estimating the $Q$-values at successive states across transitions in a trajectory. Additionally, the $CQL$ objective has the gap-expanding property because the conservatism penalty tries to push $Q$-values of different actions at a state apart (pushing up the data-action value and pushing down others to some extent). Sometimes in the case of repeated sampling of a particular transition tuple, there is a potential divergence of $Q$-values of the nearby (preceding and following) state-action pairs. This can be seen as an explosion of $Q$-values in the training metrics

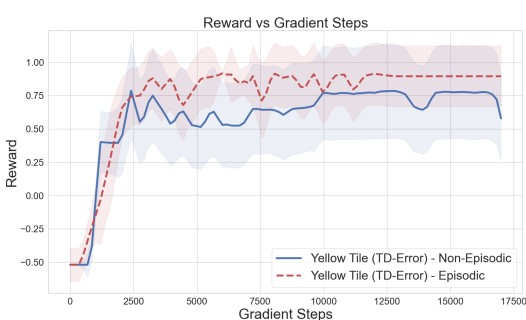

Figure 7: (**Traffic-World**) Two agents trained with and without episodic scores for TD-Error-based active sampling of causally ambigous data.

and can be partially resolved by gradient clipping. We only observed this in the case of $TD - Error$ and $Variance - data$ variants.

One experiment we tried (with some success) was that of *episodic* sampling, where we use a heuristic to convert individual transition-wise acquisition scores to scores over entire episodes (for example taking the maximum acquisition score over transitions in an episode). This kind of episodic sampling turns out to give much more stable training curves but involves additional hyperparameters and heuristics. The reasoning for its success is likely related to the motivation of algorithms like emphatic-TD [Rupam Mahmood et al., 2015], Reverse Experience Replay (RER) [Rotinov, 2019]. In these works, they propose not just prioritising transitions with high TD-error, but also increasing the priority of transitions preceding these ones in the priority queue, since these transitions will have informative TD-updates in the subsequent time-steps. We show the reward curves with episodic and without episodic active sampling of data corresponding to the experiment in Traffic-World with the yellow tile present in the data.

## I.1  Computational Cost

Figure 8 shows a scatterplot for the wallclock times to achieve highest reward across different active and uniform baselines (labelled as TD-Best, Variance-Best and Random-Best). It also plots the time needed for active sampling variants to achieve the best reward that uniform sampling achieves (denoted as Variance-par-Random and TD-par-Random in the plot).

# J  Hyper-parameters

**Active Sampling.**  We kept all the hyper-parameters the same as uniform sampling (batch size, learning rate, $\alpha$). The parameters related to active sampling are : **(1)** $n$: the number of gradient steps with stale scores we take before we recompute acquisition scores on the data. **(2)** the ensemble size which we keep constant across the active and uniform sampling variants for a fair comparison.

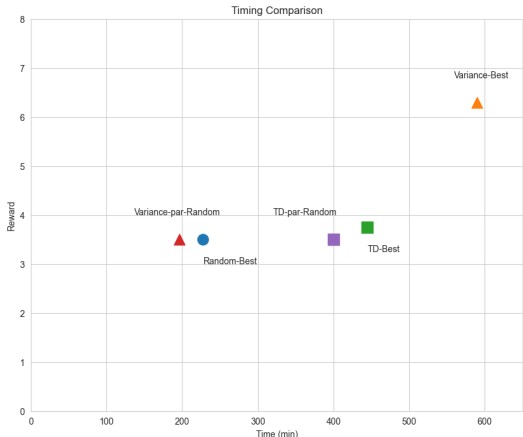

Figure 8: Timing Comparison for different sampling schemes on the Procgen-Maze benchmark plotted as reward achieved versus wallclock time in minutes.

| Hyperparameter | Value | Search |
|---|---|---|
| $\alpha$ (CQL) | 1 | 1, 4 |
| $lr$ | $5 \times 10^{-3}$ | $5 \times 10^{-4}$, $1 \times 10^{-3}$, $5 \times 10^{-3}$ |
| batch-size | 512 | 256, 512, 1024 |
| $n$ (steps before score recomputation) | 4 | 2,4,8,16 |
| gradient clipping norm | 5 | 1,3,5,7 |
| target update interval | 4 | 1, 4, 16, 32 |
| ensemble size | 3 | 3,6 |
| seeds | 7 | |

Table 1: Traffic-World experiments.

| Hyperparameter | Value | Search |
|---|---|---|
| $\alpha$ (CQL) | 1 | 1, 4 |
| $lr$ | $1 \times 10^{-3}$ | $5 \times 10^{-4}$, $1 \times 10^{-3}$, $5 \times 10^{-3}$ |
| batch-size | 2048 | 1024, 2048 |
| $n$ (steps before score recomputation) | 8 | 4, 8, 16 |
| gradient clipping norm | 5 | 1,3,5,7 |
| target update interval | 50 | 20, 50, 100 |
| ensemble size | 5 | 3, 5 |
| seeds | 9 | |

Table 2: Maze experiments.

| Hyperparameter | Value | Search |
|---|---|---|
| $\alpha$ (CQL) | 1 | 1, 4 |
| $lr$ | $5 \times 10^{-3}$ | $5 \times 10^{-4}$, $1 \times 10^{-3}$, $5 \times 10^{-3}$ |
| batch-size | 2048 | 1024, 2048 |
| $n$ (steps before score recomputation) | 8 | 8, 16 |
| gradient clipping norm | 7 | 1,3,5,7 |
| target update interval | 100 | 10, 50, 100, 1000 |
| ensemble size | 5 | 3, 5, 10 |
| seeds | 9 | |

Table 3: Enduro experiments.

