# OpenReview forum: "Can Active Sampling Reduce Causal Confusion in Offline Reinforcement Learning?"
_NeurIPS.cc/2022/Workshop/Offline_RL — Offline RL Workshop NeurIPS 2022_

### Official Review · Reviewer_yozV · 2022-10-11

**Rating:** 6
**Confidence:** 4

**Review:**

This paper proposes a way to make offline RL algorithms more robust to spurious features by modifying the replay buffer to oversample transitions where the Q functions have high uncertainty (or TD error). The proposed method increases the rewards when tested on environments where the spurious feature is changed.

Overall, I think the underlying idea makes sense, and the results are compelling. Generally, the paper is well written (see minor comments below). My one concern about the paper is whether the method works for the reason provided (causal confusion) and not for some other reason. In fact, the good performance of the TD-Error sampling suggests that the good performance of the proposed method might be more a consequence of active learning, rather than a consequence of causal confusion. I would recommend that the paper discuss this point.

Minor comments/questions
* Do all the methods in the plots use ensembles?
* L2 "imperfect spurious correlations" -- I'd recommend explaining what this means, as prior work sometimes uses "spurious" to mean different things (e.g., group shift, covariate shift)
* L5 -- I'd recommend elaborating on why spurious correlations would cause open loop performance to be different from closed loop performance (Is this really comparing online vs offline performance?)
* L8 "first study of causal confusion in offline RL" -- Doesn't [de Haan '19] also use the offline setting?
* L20 "often intractable" -- Cite.
* L23 "events" -- Cite.
* L31 "help in optimizing the training loss" -- I don't think this is true.
* L37-L47 -- This is a great explanation!
* L42 "input; OREO" --> "input. OREO"
* Contractions -- most people suggest avoiding contractions in technical writing
* L74 -- Does the bootstrap DQN paper [Osband '16] also use this strategy? If so, cite.
* Fig 2a -- Make sure this is referenced in the main text.
* Fig 2 -- Should the left/right subplots be swapped?

---

### Official Review · Reviewer_Pbig · 2022-10-20

**Rating:** 7
**Confidence:** 3

**Review:**

This paper studies causal confusion in offline RL and strategies for mitigating it. Specifically, cases where there are heavy correlations in the dataset (for example in the paper a driving dataset where 98% of the time following the car in front is a good strategy), and the agent failts on the long tail scenarios.

They propose to resolve this with active sampling, where they perform weighted sampling of transitions based on variance in the advantage function across an ensemble of Q functions. They find that this performs much better than random sampling and using TD-Error to sample in the driving example.

Overall this studies a problem relevant to workshop, and proposes a reasonable approach that seems to perform well. The paper could be improved by including experiments in a broader range of environments, including more challenging and realistic driving datasets.